# Integrating bioinformatics and machine learning to elucidate the role of protein glycosylation-related genes in the pathogenesis of diabetic kidney disease

**Ziyang Liu**[1,2], **Zengyuan Qin**[1,2], **Wenxin Bai**[1,2], **Shasha Wang**[1,2], **Chunling Huang**[1,2], **Na Li**[1,2], **Lei Yan**[1,2], **Yue Gu**[1,2]*, **Fengmin Shao**[1,2]*

**1** Department of Nephrology, Zhengzhou University People's Hospital, Henan Provincial People's Hospital, Zhengzhou, Henan, China, **2** Department of Nephrology, Henan Provincial Clinical Research Center for Kidney Disease, Henan Key Laboratory for Kidney Disease and Immunology, Henan Provincial People's Hospital, Zhengzhou University People's Hospital, Zhengzhou, Henan, China

* fengminshao@126.com (FS); guyuesunny@163.com (YG)

## Abstract

### Background

Diabetic kidney disease (DKD) is a severe global complication of diabetes, yet its molecular mechanisms remain incompletely understood. This study aimed to investigate the role of protein glycosylation in DKD pathogenesis and its association with gene expression changes, with the goal of identifying diagnostic biomarkers and personalized therapeutic targets.

### Methods

Integrated bioinformatics and machine learning approaches were applied to analyze multiple gene expression datasets. Differentially expressed glycosylation-related genes were identified, followed by unsupervised clustering to define molecular subtypes. Functional enrichment, immune cell infiltration analysis, and machine learning algorithms (including feature selection for hub genes) were employed. qPCR validation was performed on clinical DKD and normal kidney tissues, and ROC curves were generated to assess diagnostic potential.

### Results

Unsupervised clustering of glycosylation-related genes revealed two distinct DKD molecular subtypes with differential pathway activation (e.g., extracellular matrix remodeling) and immune infiltration patterns. Six hub genes (S100A12, EXT1, SBSPON, ADAMTS1, FMOD, SPTB) were identified as critical to DKD pathogenesis through machine learning. Immune infiltration analysis showed significant differences in macrophage and neutrophil activity between DKD and controls and

**Data availability statement:** Three DKD-related datasets, GSE96804 (41 cases of DKD, 20 cases of normal renal tissue), GSE104948-GPL22945 (7 cases of DKD, 18 cases of normal renal tissue), and GSE30528 (9 cases of DKD, 13 cases of normal renal tissue), were downloaded from the Gene Expression Omnibus (GEO) database (www.ncbi.nlm.nih.gov/geo/). Pathways and gene sets related to glycosylation were obtained from the MSigDB database (http://www.gsea-msigdb.org/gsea/msigdb/).

**Funding:** Specific funding sources are as follows: Henan Province Major Science and Technology Project (241100310100), Henan Province Clinical Research Doctor Training Special Project (D20240006) and Medical Science and Technology Attack Plan Project of Henan Province (SBGJ202301001). The funders played no role in the study design, data collection and analysis, decision to publish, or preparation of the manuscript. Their contributions were limited to providing financial support for the research, which was managed by the authors.

**Competing interests:** The authors have declared that no competing interests exist.

Immunohistochemical results confirmed the occurrence of immune infiltration. qPCR validation confirmed dysregulation of hub genes in DKD tissues compared to normal samples. ROC analysis demonstrated high diagnostic accuracy for these genes.

## Conclusions

This study highlights abnormal protein glycosylation as a key player in DKD and identifies six hub genes with potential as diagnostic biomarkers. The molecular subtypes and immune infiltration patterns provide insights into disease heterogeneity, paving the way for personalized therapies. Future studies should validate these findings in larger cohorts with explicit sample sizes to strengthen clinical applicability.

## Introduction

Diabetic kidney disease (DKD) is a chronic kidney disease caused by diabetes mellitus (DM) and is one of the most common complications of DM. Of the estimated 400 million people with type 2 diabetes worldwide, about half will progress to DKD, and nearly one-third of people with type 1 diabetes will develop this complication [1]. The main clinical manifestations include renal dysfunction, proteinuria, and hypertension. In severe cases, the condition can lead to end-stage renal disease, requiring dialysis or kidney transplantation to sustain life. As a serious complication, DKD not only significantly impacts the patient's health and quality of life but also imposes a heavy burden on families and society [2]. DKD is characterized by an insidious onset. In the early stages, there may be no obvious symptoms or only microalbuminuria. However, when massive proteinuria or other obvious symptoms occur, the disease usually has progressed to an advanced stage [3]. Therefore, in-depth research on the pathogenesis of DKD, searching for molecules that play important roles in the disease's development, and exploring specific and sensitive biomarkers are of great significance for its diagnosis and treatment.

Protein glycosylation is a biological process by which glycosyltransferases add sugar groups to protein molecules through covalent bonds, forming glycoproteins. Protein glycosylation, one of the most common post-translational modifications (PTMs), is mainly divided into N-linked and O-linked glycosylation. It plays a crucial role in various biological processes, including protein folding and stability, cell signaling, immune response, and metabolic regulation [4]. For example, studies by Ravidà et al. have shown different glycosylation patterns in DKD rats' kidney proteins, and significant disease-related differences in glycoprotein expression among five lectins (UEAI, PHA-E, GSI, PNA, RCA) were identified [5]. A prospective cohort study involving 688 patients with type 2 diabetes and 134 controls without diabetes and chronic kidney disease found that urinary glycan profiling is associated with renal outcomes in patients with type 2 diabetes [6].

N-linked glycosylation mainly occurs in the endoplasmic reticulum and Golgi apparatus. Glycans are linked to asparagine (Asn) residues of proteins through N-acetylglucosamine (GlcNAc), playing important roles in protein folding, stability

and transport. Bermingham et al. screened 818 patients with severe annual eGFR decline and analyzed their total N-glycosylation profiles as well as IgG-specific N-glycosylation profiles. They found a correlation between HbA1c levels and alterations in serum N-glycosylation profiles in individuals with type 1 diabetes mellitus and aberrant N-glycosylation has been associated with albumin-to-creatinine ratio (ACR) and estimated glomerular filtration rate (eGFR) decline [7]. Xu et al. discovered that ENTPD5 regulates renal tubular cell proliferation and apoptosis by mediating protein N-glycosylation in the endoplasmic reticulum, as demonstrated through studies on human renal biopsy samples and db/db mice. ENTPD5-regulated N-glycosylation of proteins in the ER to promote cell proliferation in the early stage of DKD, and continuous hyperglycemia activated the hexosamine biosynthesis pathway (HBP) to increase the level of UDP-GlcNAc, which driving a feedback mechanism that inhibited transcription factor SP1 activity to down-regulate ENTPD5 expression in the late stage of DKD.[8]. Delička Deličković et al. identified significant differences in N-glycans between healthy and sclerotic glomeruli in FFPE sections of renal biopsy samples from five DKD patients and three healthy adults [9].

O-linked glycosylation typically occurs in the Golgi apparatus, with O-GlcNAcylation being the most common and studied type. This modification involves linking a sugar group to a protein's serine (Ser) or threonine (Thr) residue via N-acetylglucosamine (GlcNAc), influencing cell transcription regulation, signal transduction and metabolic regulation. Degrell et al. found increased O-GlcNAcylation levels in the glomeruli and renal tubules of DKD patients when analyzing immunohistochemical samples from renal biopsies [10]. Goldberg et al. demonstrated that O-GlcNAcylation in high glucose-cultured glomerular mesangial cells activates diabetic kidney fibrosis [11]. Park et al. investigated the effects of high glucose on O-GlcNAcylated carbohydrate response element-binding protein (ChREBP) in streptozotocin-induced diabetic rat mesangial cells, revealing that elevated glucose induces ChREBP upregulation, resulting in lipid accumulation and increased fibrin expression in mesangial cells [12]. Gellai et al. studied high glucose-cultured renal tubular epithelial cells and a diabetic rat model, finding that high glucose-induced O-GlcNAcylation promoted renal fibrosis in DKD by inhibiting Akt/eNOS phosphorylation and inducing HSP72 expression [13]. Sugahara et al. found that a high-fat diet increases O-GlcNAcylation levels in renal tubular epithelial cells of obese diabetic mice, maintaining cellular lipid metabolism and protecting against diabetes-related renal lipotoxicity [14].

The pathogenesis of DKD is complex and has not yet been fully elucidated. Protein glycosylation maintains the structural integrity and functional competence of glomerular and tubular cells by regulating protein functionality and cell signaling pathways. Moreover, it also plays an important role in the pathogenesis of DKD by mediating inflammatory response and oxidative stress [15–17]. Machine learning is a method for predicting or making decisions about future unidentified data by acquiring knowledge from historical or current data. The use of machine learning technology to screen biomarkers can effectively assist in accurate disease prediction and patient stratification, advancing precision medicine and offering extensive prospects for personalized treatment options [18,19]. The present study utilized bioinformatics integrated with machine learning to identify the genes and their regulatory networks associated with protein glycosylation, which plays a pivotal role in DKD. Additionally, we constructed and evaluated a clinical risk prediction model, providing valuable insights for the discovery of novel biomarkers and therapeutic targets.

## Materials and methods

### 1.1. Data acquisition and pre-processing

Three DKD-related datasets, GSE96804 (41 cases of DKD, 20 cases of normal renal tissue), GSE104948-GPL22945 (7 cases of DKD, 18 cases of normal renal tissue), and GSE30528 (9 cases of DKD, 13 cases of normal renal tissue), were downloaded from the Gene Expression Omnibus (GEO) database (www.ncbi.nlm.nih.gov/geo/). Pathways and gene sets related to glycosylation were obtained from the MSigDB database (http://www.gsea-msigdb.org/gsea/msigdb/). To remove batch effects due to experimental errors, the R packages "limma" and "sva" were used for data pre-processing, allowing the combined dataset to focus more on biological differences. Principal component analysis (PCA) scatter plots before and

after merging were drawn using the R packages "FactoMineR" and "factoextra," respectively. Subsequently, the R package "preprocessCore" was used for data homogenization to reduce the adverse effects of outlier data on downstream analysis.

## 1.2. Identification of DEGs, gly-DEGs, and GO, KEGG analysis

The R package "limma" was used for differential expression analysis, and differentially expressed genes (DEGs) were screened according to the criteria of adj. P. value < 0.05 and |logFC| > 0.5. The differentially expressed glycosylation modification-related genes (gly-DEGs) were obtained from the intersection of the selected DEGs and the glycosylation-related gene sets. Next, the "ggpubr" and "ggthemes" packages were used to draw volcano plots, and the "pheatmap" package was used to generate heat maps to show the significant changes in gene expression. Then, the R package "clusterProfiler" was used on DEGs and gly-DEGs for Gene Ontology (GO) annotation and Kyoto Encyclopedia of Genes and Genomes (KEGG) pathway enrichment analysis, with results visualized using histograms and bubble charts.

## 1.3. Sample clustering and subtyping

The R package "ConsensusClusterPlus" was utilized to perform unsupervised cluster analysis of the samples based on gly-DEGs, resulting in the identification of two distinct subtypes that best classify the samples. The classification results were visualized using the "pheatmap" package, which employs heat maps to demonstrate the clinical features of each type and the relationship between gene expression patterns. To further observe the similarities and differences between the subtypes, a PCA scatter plot was generated. Subsequently, the differential genes between the different subtypes were analyzed, and a volcano plot was drawn.

## 1.4. GSVA, GO, and KEGG enrichment analysis after sample clustering and subtyping

The HALLMARK, KEGG, and Reactome pathway datasets were downloaded from the Msigdb database. Based on the typing results in Section 1.3, pathway scores for each sample were calculated using the R package "GSVA" to evaluate the differences in pathway activity between the different subtypes. Then, the R package "pheatmap" was used to draw a heatmap to visually show the differences in these pathways between the two groups of samples. Based on the pathway scoring results, the differences between the subtypes were analyzed, and the DEGs were further analyzed for functional enrichment. The R package "clusterProfiler" was used to perform GO annotation and KEGG pathway enrichment analysis on the DEGs to explore their significance in the biological pathways.

## 1.5. Identification of hub genes by machine learning

LASSO regression analysis was performed on gly-DEGs using the R package "glmnet" to screen out genes with significant features. To minimize overfitting and enhance model generalizability, all machine learning models (LASSO, random forest, and SVM) were trained using 10-fold cross-validation, where the dataset was repeatedly partitioned into 10 subsets for training and validation. Then, the "randomForest" package was used to build a random forest model, and the top 10 genes were selected based on their importance ranking. The support vector machine (SVM) model was then constructed using the "kernlab" package to further screen the relevant genes. Finally, by integrating the gene feature sets selected by the three machine learning models, the hub genes were obtained by intersection, and the correlation analysis of these genes was performed. The receiver operating characteristic (ROC) curve was drawn using the "pROC" package to evaluate their diagnostic value. The area under the ROC curve (AUC) was calculated, and an AUC greater than 0.7 indicated that these genes had good diagnostic performance.

## 1.6. Establishment of DKD model

C57BKS db/db male mice (8 weeks old, n = 3) were selected as the experimental group, while age- and sex-matched C57BKS db/m male mice (n = 3) served as the control group. Mice were housed in a specific pathogen-free (SPF)

environment with a 12-hour light/dark cycle, controlled temperature (22±2°C), and humidity (55±5%). |The experimental group was fed a high-fat diet (HFD, 60 kcal% fat) to accelerate diabetes progression, while the control group received a standard chow diet. Body weight, non-fasting blood glucose levels (via tail vein blood using a glucometer), and urinary protein content (measured by dipstick) were monitored weekly to assess disease progression.

Anesthesia and Analgesia. At 16 weeks, all mice underwent terminal procedures under deep anesthesia to minimize pain and distress. Anesthesia was induced via intraperitoneal injection of sodium pentobarbital (150 mg/kg, dissolved in sterile saline), with additional supplemental doses administered as needed to maintain surgical plane anesthesia (assessed by lack of pedal reflex and reduced respiratory rate). Throughout the procedure, vital signs (respiratory rate, heart rate) were visually monitored to ensure adequate anesthesia depth. No postoperative analgesia was required as procedures were terminal.

Euthanasia and Tissue Collection. Euthanasia was performed by overdose of sodium pentobarbital (200 mg/kg, intraperitoneal injection), a method recommended by the American Veterinary Medical Association (AVMA) Guidelines for Euthanasia to ensure rapid and painless death. Following confirmation of absent corneal reflexes and cessation of breathing, cardiac blood was collected via puncture of the right ventricle, and kidneys were rapidly excised. Tissues were immediately processed for subsequent analyses (e.g., qPCR, immunohistochemistry).

Efforts to Alleviate Suffering. All efforts were made to minimize animal discomfort: Humane Endpoints: Mice were monitored daily for signs of distress (e.g., hunched posture, reduced activity, abnormal respiration). Any animal exhibiting severe morbidity (e.g., blood glucose >300 mg/dL combined with weight loss >20%) would have been humanely euthanized prior to the scheduled endpoint. 3Rs Principles: The study used the minimum number of animals required to achieve statistical validity (n=3 per group) and optimized procedures to reduce pain, such as using intraperitoneal anesthesia to avoid tissue trauma from inhalation methods. Training and Care: All personnel handling animals were trained in aseptic technique and mouse husbandry to ensure gentle handling and minimize stress.

The protocol was approved by the Experimental Animal Ethics Committee of Zhengzhou University (Approval No. ZZU-LAC20240220[01]) and strictly adhered to national and institutional guidelines for the care and use of laboratory animals.

## 1.7. Validation of hub gene expression by qPCR

Total RNA was extracted from renal tissues of DKD and control groups using the FastPure Complex Tissue/Cell Total RNA Isolation Kit (Vazyme, cat# RC113−01) following the manufacturer's protocol. cDNA synthesis was performed with 1 µg RNA using the HiScript III RT SuperMix (Vazyme, cat# R323-01). Specific primers for hub genes (sequences listed in Primer List for qPCR Validation) were synthesized by Sangon Biotech (Shanghai, China). qPCR was conducted using ChamQ Universal SYBR qPCR Master Mix (Vazyme, cat# Q711-02) and thermal cycling conditions were as follows: 95°C for 3 min, followed by 40 cycles of 95°C for 15 sec and 60°C for 30 sec. GAPDH was used as the internal reference gene, and relative expression levels were calculated using the $2-\Delta\Delta Ct$ method. Statistical analysis was performed using Graph-Pad Prism 9.0 to compare gene expression differences between groups.

| Gene | Forward Primer (5'→3') | Reverse Primer (5'→3') |
|---|---|---|
| EXT1 | GGCAAAAGCACAGGATTCGC | CTGCAAAGCCTCAGGAATCTG |
| S100A12 | GGATGCTAATCAAGATGAACAGGTC | ACTCTTTGTGGGTGTGGTAATGG |
| SBSPON | GTGTTCTGCGACCAAGCCTGT | GAGCACTGTTCTACTCGAGCCA |
| ADAMTS1 | GAAGGCAAACGAGTCCGCTACA | TTGGGTGTCACTCTAGAGTGG |
| FMOD | CCAACACCCTTCAACTCAGAG | GTGCAGAAGCTGCTGATGGAGA |
| SPTB | TGAAGCCATCCTCAGCAACAG | GGACTCAGGATCTTGTCTCGGT |
| GAPDH | TGCACCACCAACTGCTTAG | GGATGCAGGGATGATGTTTC |

Primer List for qPCR Validation

## 1.8. Immunohistochemistry

Kidney tissues were fixed in 4% paraformaldehyde (Biosharp, cat# BL539A) for 24 hours, embedded in paraffin, and sectioned at 4 μm thickness. Deparaffinization was performed using xylene, followed by rehydration through a graded ethanol series. Antigen retrieval was conducted with EDTA buffer (pH 9.0) at 95°C for 20 minutes. Endogenous peroxidase activity was blocked with 3% hydrogen peroxide for 10 minutes, and non-specific binding was minimized using 5% normal goat serum (Invitrogen, cat# 31872) for 30 minutes. Sections were incubated overnight at 4°C with primary antibodies against CD45 (Abcam, cat# ab10558, 1:200 dilution). After washing, sections were treated with HRP-conjugated secondary antibody (Servicebio, cat# GB23303, 1:500 dilution) for 30 minutes at room temperature. Signal detection was performed using DAB substrate (Servicebio, cat# G1212-200T), counterstain with hematoxylin, dehydrate, clear in xylene, and mount. Analyze under a light microscope.

## 1.9. Enrichment analysis and immunoassay of hub genes using GSEA

Perform correlation analysis between the 6 hub genes and the gene expression data of the entire genome, and display the expression patterns of the top 50 genes positively correlated with each hub gene using a heatmap. Based on the results of the correlation analysis, the R package "clusterProfiler" was used to perform GSEA on these six hub genes, and the top 20 results that were significantly enriched in Reactome pathways were displayed. Then, the ssGSEA function of the R package "GSVA" was used to evaluate the degree of immune cell infiltration in the samples. The "GSVA", "GSEA-Base", and "ggplot2" packages were combined to visualize the results.

## 1.10. Prediction of transcription factors and MicroRNAs associated with hub genes

The RegNetwork database (https://regnetworkweb.org/) was used to predict the upstream transcription factors and microRNAs regulating the hub genes, exploring the potential roles of these molecules in gene regulation. Then, Cytoscape software was used to construct gene regulatory networks, which intuitively show the core interactions between the hub genes and their upstream regulatory factors.

# Results

## 2.1. Identification of DEGs

The datasets GSE96804, GSE104948-GPL22945, and GSE30528 were merged to obtain a dataset containing 11,348 genes and 108 samples. Then the batch effect was removed, and data homogenization was performed to eliminate the influence of outlier samples and ensure that the similarity between samples is more intuitive (Fig 1A: before homogenization, Fig 1B: after homogenization). On this basis, differential gene analysis was performed. According to the criteria of adj. P. value < 0.05 and |logFC| > 0.5, 278 genes were up-regulated and 309 genes were down-regulated in DKD compared with normal kidney tissue (Fig 1C and 1D).

## 2.2. GO and KEGG Enrichment Analysis of DEGs

To determine the biological functions of the DEGs, GO and KEGG pathway enrichment analyses were performed. GO analysis revealed that DEGs were significantly enriched in biological processes such as urinary and reproductive system development, leukocyte migration, kidney system development, cell chemotaxis, and wound healing (Fig 2A). These processes may be associated with tissue damage, chronic inflammation, and impaired renal function in DKD. The significant enrichment of cellular components, such as the extracellular matrix containing collagen, suggests that these genes may be involved in the formation of glomerulosclerosis and fibrosis, which are important pathological features of DKD (Fig 2B). The enrichment in molecular functions related to receptor-ligand activity, signal receptor

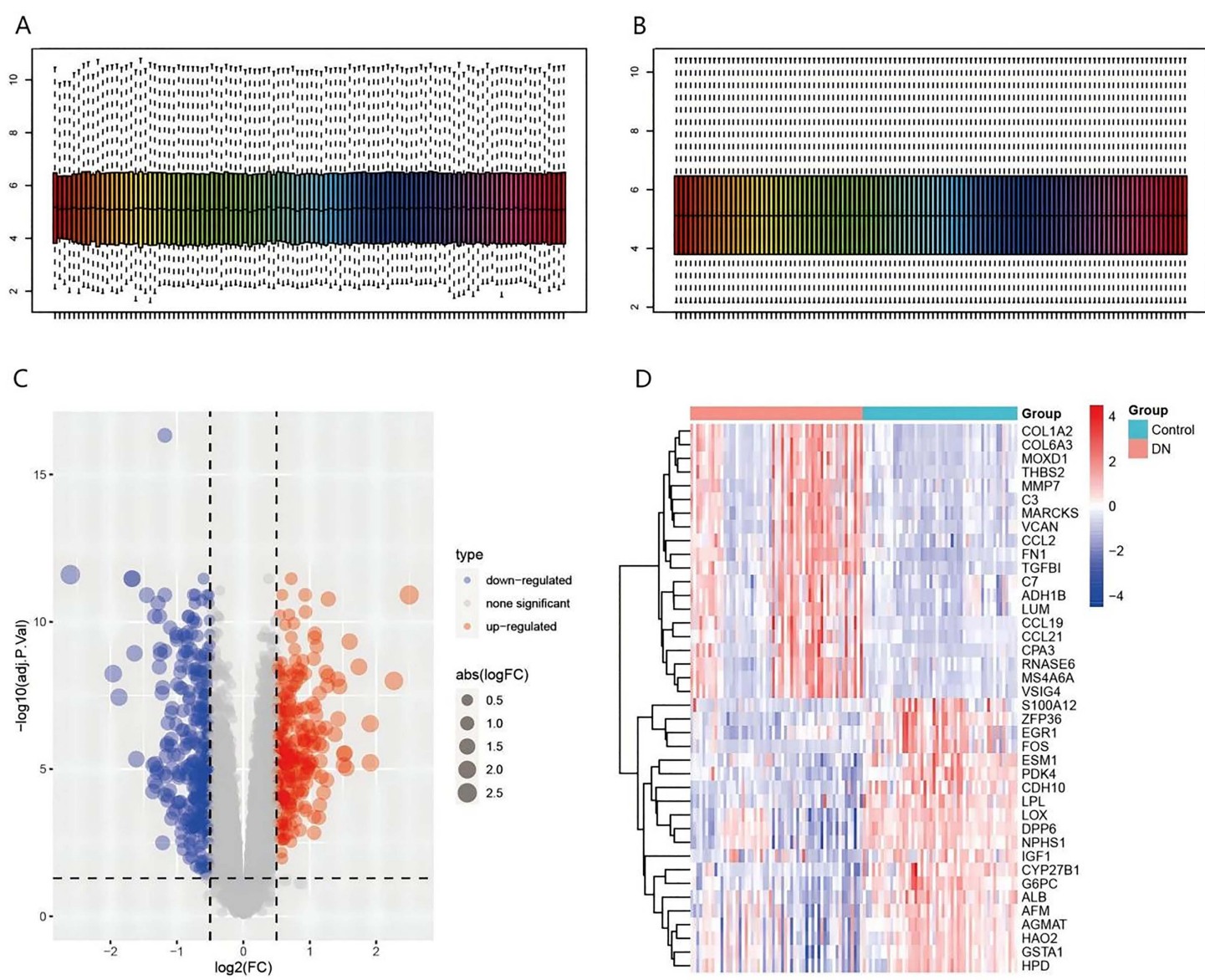

**Fig 1. Data pre-processing and identification of DEGs.** Data normalization mitigated outlier effects (A-B), showing pre- (A) and post-normalization (B) data. Differential expression analysis (C-D) using an adjusted p-value <0.05 and |logFC| >0.5 revealed 278 up-regulated and 309 down-regulated genes. The volcano plot (C) depicts differentially expressed genes (blue: down-regulated; red: up-regulated; gray: non-significant). The heatmap (D) visualizes differential expression between patients and healthy controls (red: higher expression; blue: lower expression).

activator activity, extracellular matrix structural composition, glycosaminoglycan binding, and sulfur compound binding may indicate abnormalities in cell signaling and regulation of the extracellular matrix (Fig 2C). KEGG analysis showed a significant enrichment in DEGs in PI3K-Akt signaling pathways, cytokine-cytokine receptor interaction, and MAPK signaling pathways (Fig 2D). These pathways are closely associated with cell proliferation, survival, metabolism, and inflammatory responses, and their abnormal activation may be a key factor in the pathological progression of DKD.

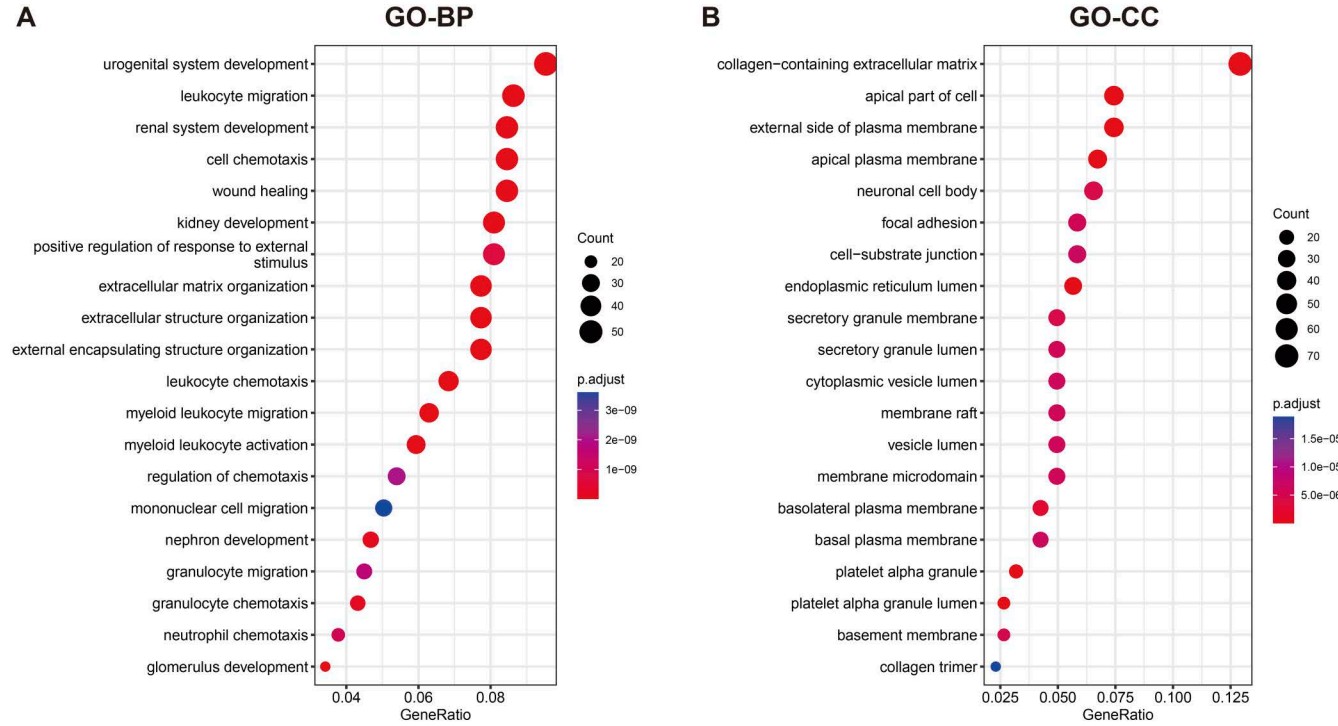

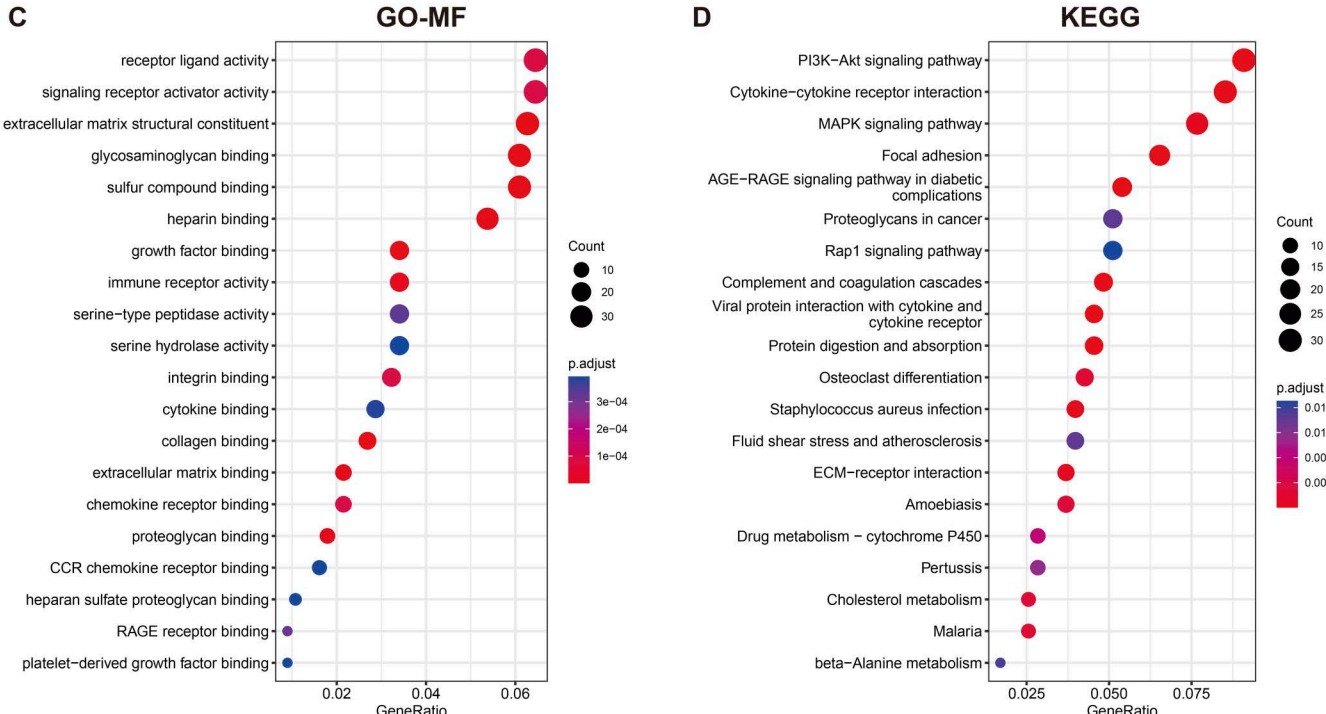

**Fig 2. GO annotation and KEGG pathway enrichment analysis of DEGs.** GO annotations highlight enrichment in biological processes (BP) (A), cellular components (CC) (B), and molecular functions (MF) (C). KEGG pathway enrichment is presented in (D), ranked by GeneRatio. Circle size indicates the number of enriched genes, and color represents p-values.

## 2.3. Identification of gly-DEGs

The intersection of the DEGs obtained in Section 2.1 and the glycosylation gene set yielded gly-DEGs, including 13 upregulated genes (THBS2, EXT1, LUM, FMOD, ADAMTS1, etc.) (Fig 3A) and 8 downregulated genes (SPTB, SEMA5A, S100A12, THSD7A, B3GALT2, etc.) (Fig 3B). The differential expression of these 21 gly-DEGs between patients and healthy controls was shown using volcano plots (Fig 4A) and heatmaps (Fig 4B). Boxplots were used to display their expression levels in patients and healthy controls (Fig 4C).

## 2.4. GO and KEGG enrichment analysis of Gly-DEGs

To further determine the biological functions of gly-DEGs, GO and KEGG pathway enrichment analyses were performed. According to the GO analysis, gly-DEGs were significantly enriched in extracellular matrix organization, extracellular structure organization, and other biological processes such as encapsulation structure and significant enriched in cellular components such as extracellular matrix containing collagen. Genes enriched in molecular functions such as glycosaminoglycan binding and sulfur compound binding may influence the kidney repair and regeneration ability of patients with diabetes (Fig 3C).

   KEGG analysis showed that gly-DEGs were significantly enriched in pathways such as phagosome formation, other types of O-glycan biosynthesis, mucin-type O-glycan biosynthesis, glycolipid biosynthesis (lactose and new lactose series), and glycosaminoglycan biosynthesis (heparan sulfate/heparin pathways) (Fig 3D). The correspondence between the top 5 KEGG pathways and gly-DEGs is shown in Fig 3E.

## 2.5. Sample clustering and subtyping analysis

Unsupervised clustering was performed on the kidney samples in the dataset based on the gly-DEGs. The results showed that when the k value was set to 2, the consensus clustering matrix exhibited the most distinct discrimination, a stable number of clusters, and the highest consistency scores among different subtypes (Fig 5A). The PCA scatter plot also demonstrated significant differences between the two subtypes (Fig 6A). To further explore the molecular characteristics between subtypes, we comprehensively evaluated the expression differences of gly-DEGs in each subtype (Fig 5B). The results showed that the expression levels of EXT1, LUM, FMOD, ADAMTS1, GALNT1, LGALS3, VCAN, PRELP, ADAMTSL3, CTSC, and TUBB2B in the 21 gly-DEGs were significantly increased in DKD compared with normal kidney tissue, while the expressions of SPTB, SEMA5A, THSD7A, B3GALT2, and UMOD were significantly decreased. Heat maps illustrated the relationship between clinical features, gene expression, and subtype (Fig 5C).

## 2.6. GSVA, GO, and KEGG enrichment analysis after sample clustering and subtyping

HALLMARK pathways, KEGG pathways, and Reactome pathways were downloaded from the Msigdb database. The GSVA package in R was used to score these pathways. Comparing the differences in pathways between Type 1 and Type 2, the results indicate clear disparities between the two types. To visualize these differences, heatmaps of pathway scores between the two groups were plotted separately (Fig 5D–F).

   To analyze the differences between classifications (Fig 6B), GO and KEGG enrichment analyses were performed on DEGs (Fig 6C–D). GO analysis showed DEGs significant enrichment in to wound healing, WBC migration, and other bio-logical processes, as well as in cellular components such as the collagen-containing extracellular matrix. KEGG analysis showed significant enrichment in DEGs within the PI3K-Akt signaling pathways, focal adhesion, MAPK signaling pathways, and cytokine-cytokine receptor interaction pathways.

## 2.7. Identification of hub genes by machine learning

In this study, machine learning methods were used to select hub genes from 21 gly-DEGs. Firstly, LASSO regression was employed to identify 12 genes (Fig 7A), which utilized an L1 regularization penalty to shrink some insignificant gene

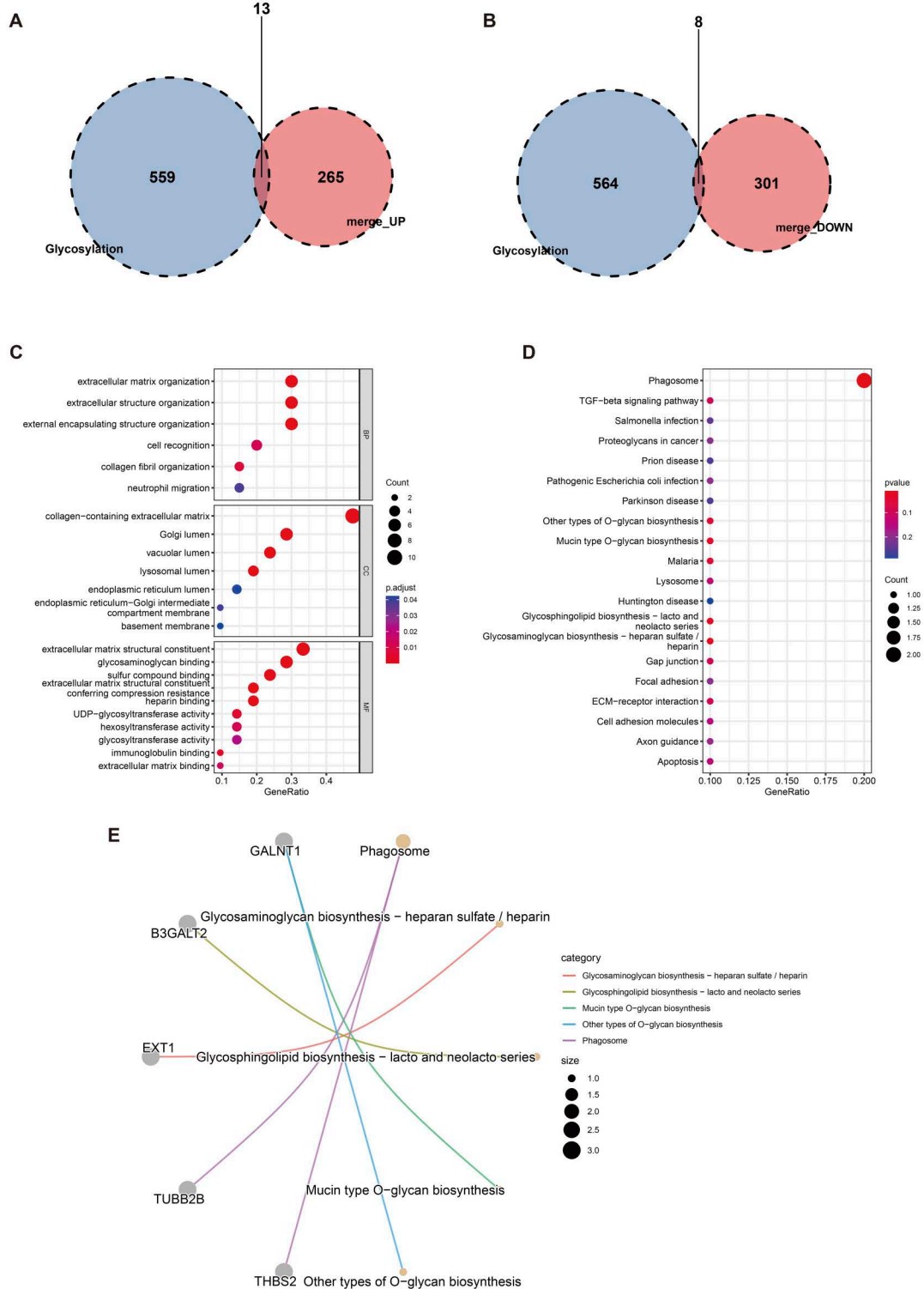

**Fig 3. Identification of gly-DEGs.** (A) displays the 13 up-regulated genes overlapping with the glycosylation gene set, while (B) shows the 8 down-regulated overlapping genes. GO annotation (C) of these glycosylation-related genes highlights enrichment in BP, CC, and MF categories. KEGG pathway enrichment analysis is presented in (D), and (E) visualizes the relationships between the top 5 enriched pathways and their associated genes, with different colored lines representing different pathways.

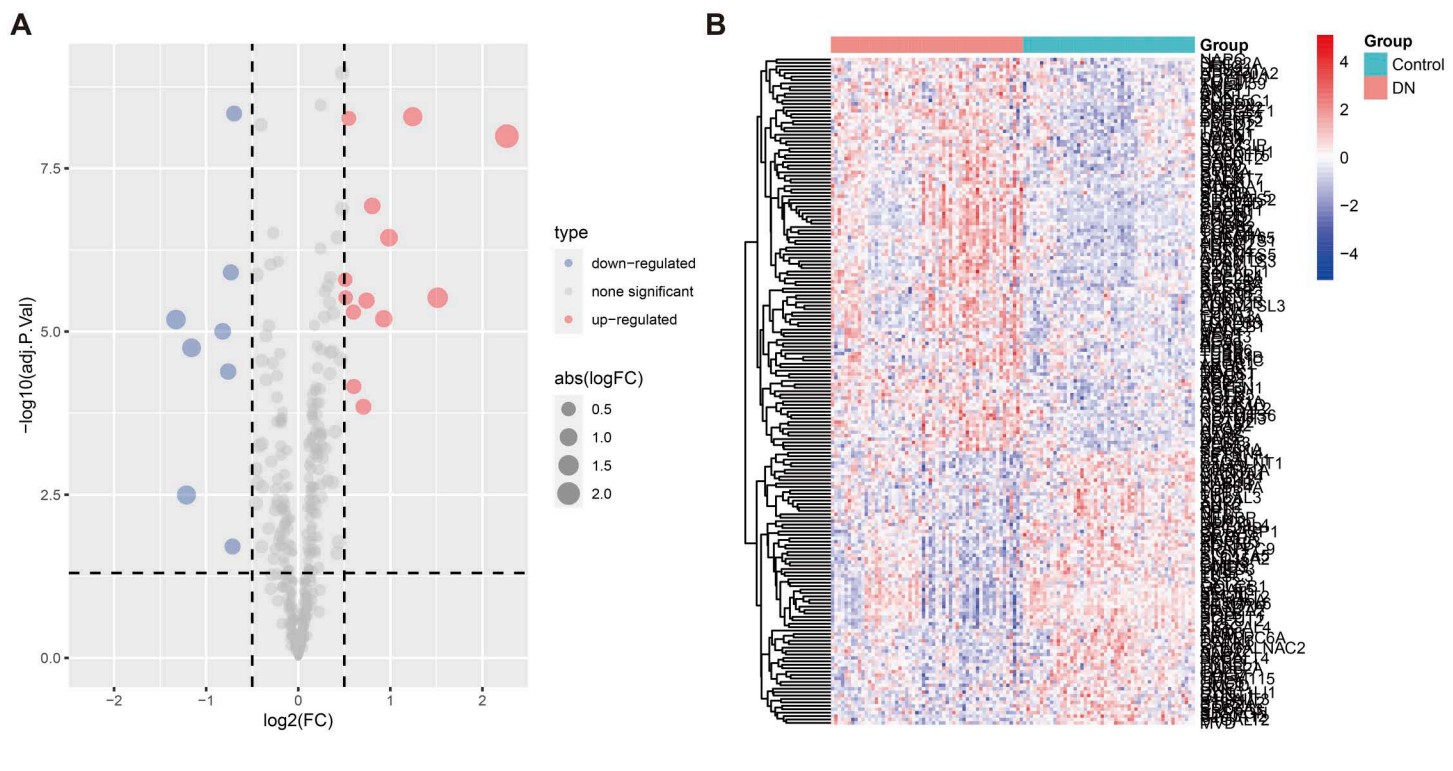

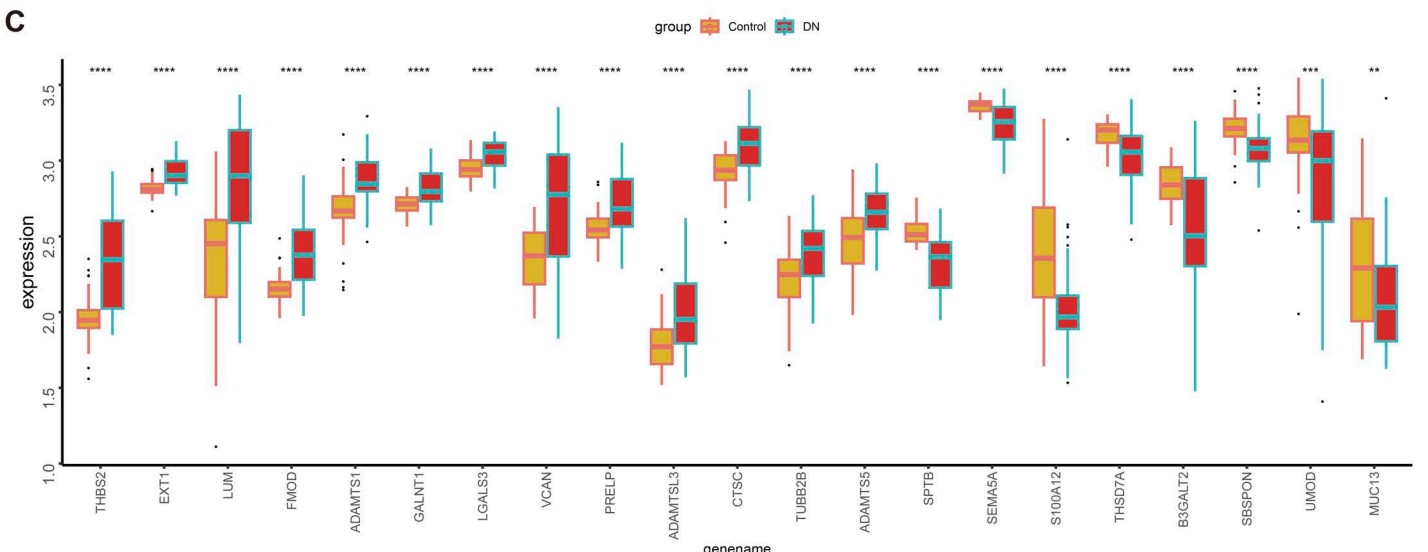

**Fig 4. Enrichment results of gly-DEGs.** Differential expression of 21 genes between patients and healthy controls is visualized in (C) a volcano plot and (D) a heatmap. (E) Box plots depict the expression levels of these 21 genes in both groups (ns: p > 0.05, *: p < 0.05, **: p < 0.01, ***: p < 0.001).

coefficients to zero and select the most influential ones. Secondly, a random forest model was applied to rank the importance of genes, and the top 10 genes were selected (Fig 7B). The random forest model evaluated the importance of each gene by constructing multiple decision trees. Finally, a support vector machine (SVM) identified 13 genes (Fig 7C) by

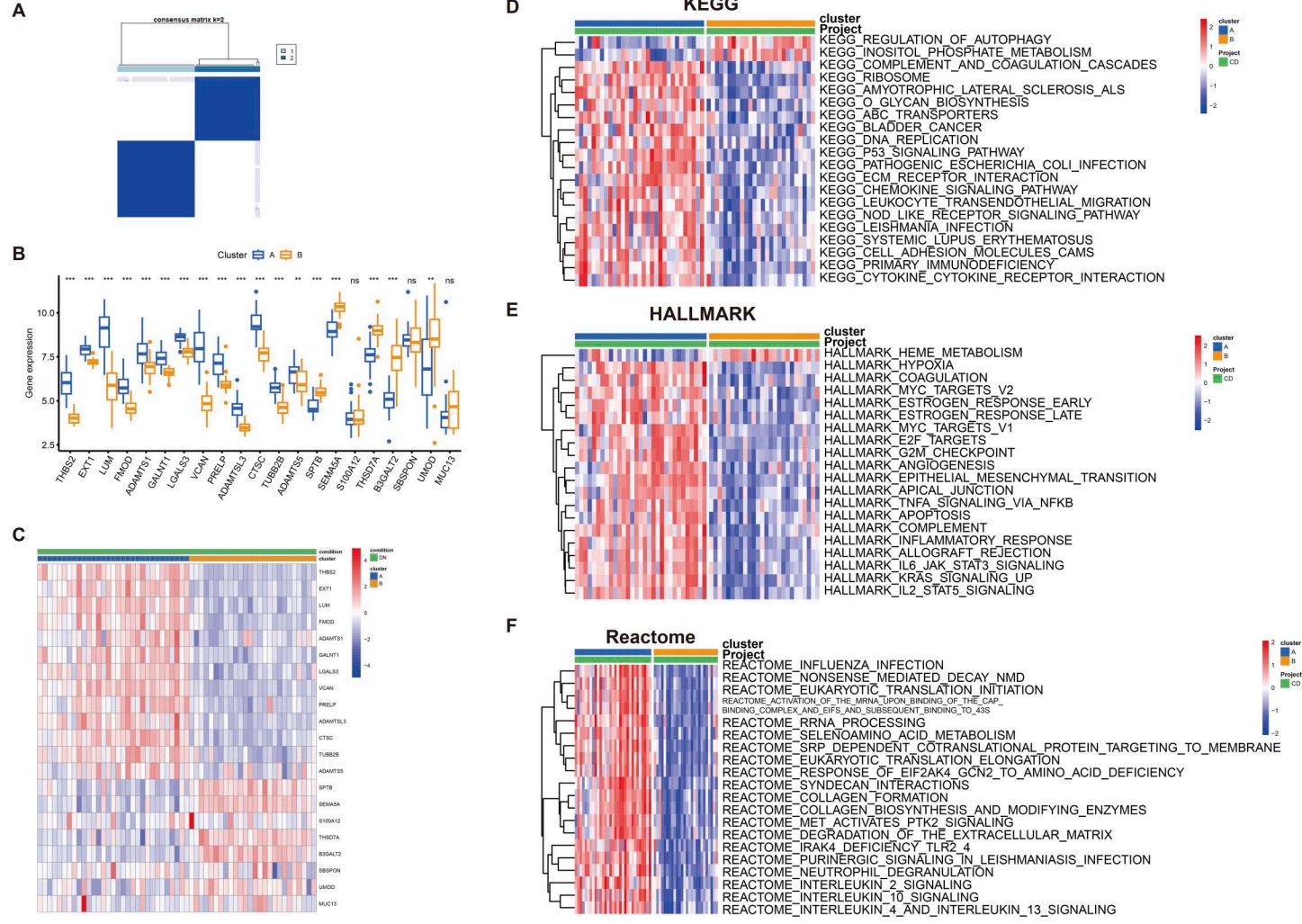

**Fig 5. Sample clustering and subtyping analysis.** A unsupervised cluster typing based on the above 21 differential genes showed that type 2 was the best. B shows the differential expression of 21 differential genes among different types, ns p > 0.05,*p < 0.05,**p < 0.01,***p < 0.001; C Heat map shows the relationship between clinical features, gene expression and classification. D-F performed pathway scores for HALLMARK pathway, KEGG pathway and Reactome pathway. Heat map drawing comparison between the two groups respectively, comparing the difference between the pathways between type 2. The results showed that there were significant differences in the pathways between the two types.

searching for an optimal classification plane in high-dimensional space that could best differentiate DKD samples. By integrating the results from these three methods and taking their intersection, six hub genes were obtained: S100A12, EXT1, SBSPON, ADAMTS1, FMOD, and SPTB (Fig 7D). The mechanisms and potential clinical applications of these genes in DKD warrant further investigation to validate their potential as early diagnostic biomarkers.

## 2.8. Relative expression of hub genes in DKD mouse model

The DKD animal model was established using C57BKS db/db mice, with db/m mice as controls. At 16 weeks, blood glucose levels and the urinary protein-to-creatinine ratio (UPCR) were measured, and paraffin sections of kidney tissue were examined to confirm successful modeling. Subsequently, the relative expression of hub genes in the kidney tissues of the experimental and control groups was compared using qPCR. The results showed that the expression of EXT1 and

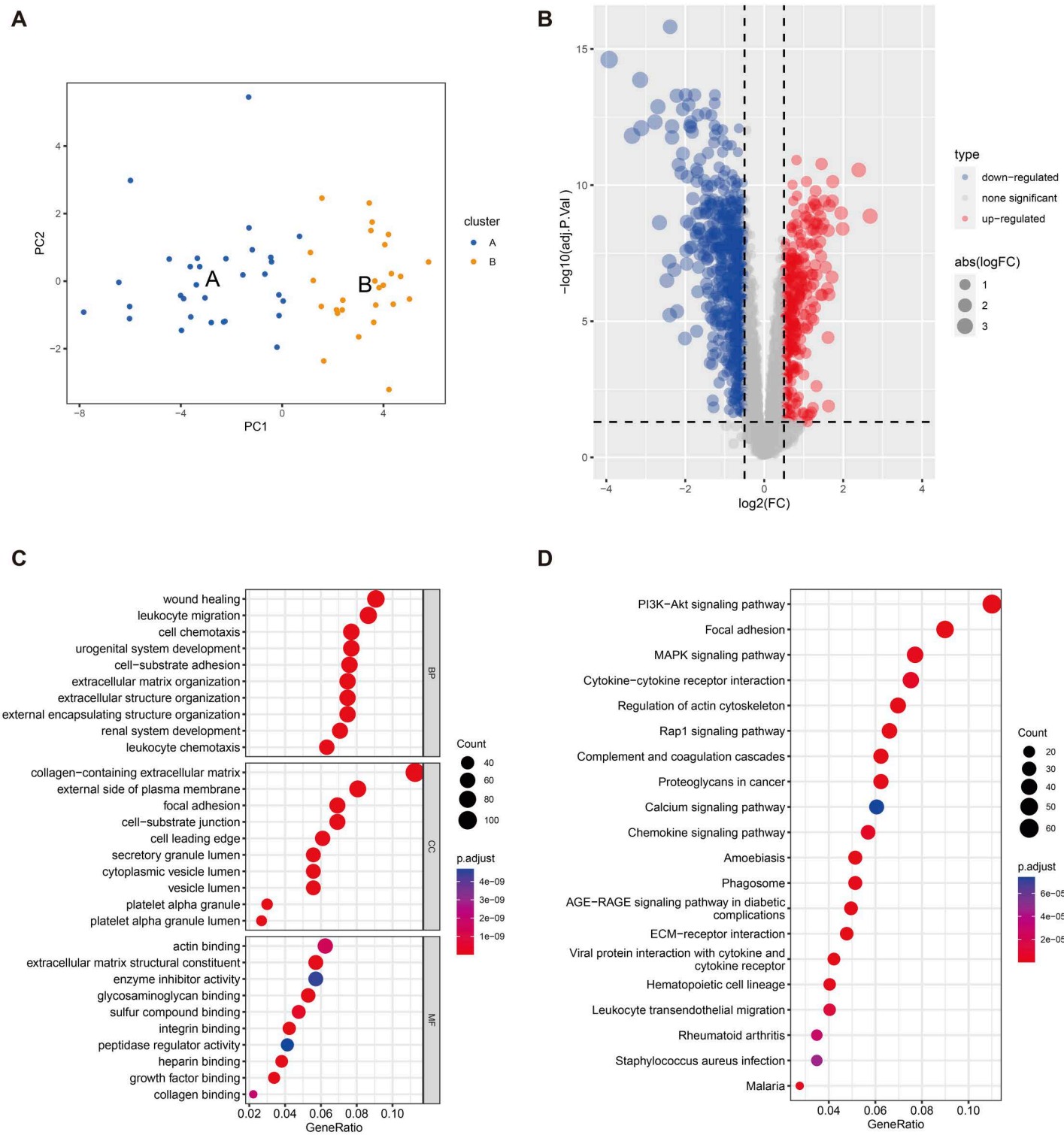

**Fig 6. GSVA, GO, and KEGG Enrichment Analysis After Sample Clustering and Subtyping.** A PCA diagram shows different parting sample distribution; B Volcano plot shows the difference analysis between different types. C-D performed enrichment analysis of differential genes. C GO annotations, respectively show in Biological processes (in the Process, BP), cell composition (Cellular Component, CC) and Function of molecules (Molecular Function, MF) D KEGG results above are sorted by in the path of GeneRatio size, the size of the circle represents the number of gene enrichment, color represents P values.

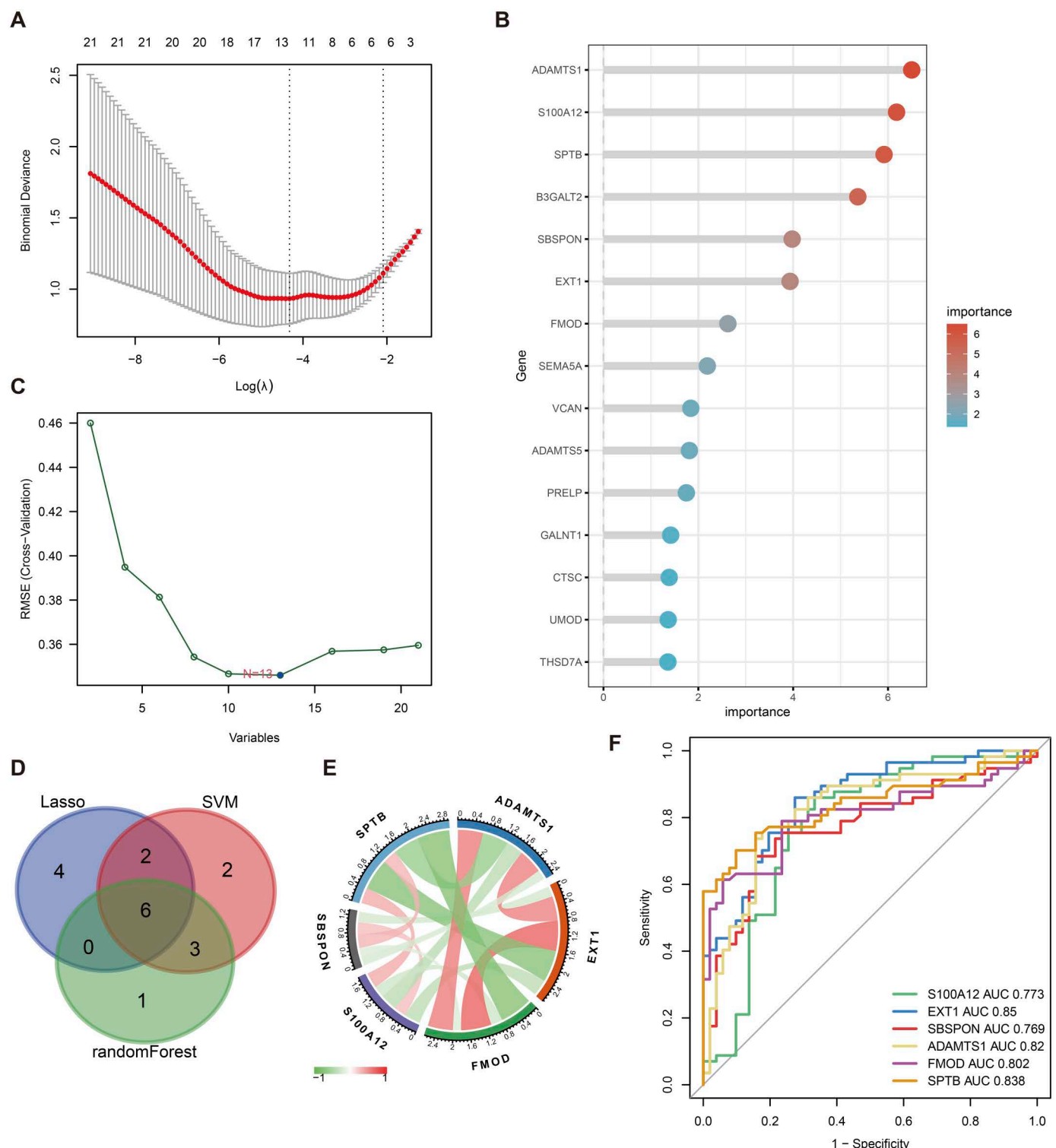

**Fig 7. Machine learning from 21 gly - DEGs filter core gene.** A LASSO regression filter to 12 genes; B random forest according to the importance to choose 10 genes; C the SVM support vector machine (SVM) screening to 13 genes; Six core D intersection control genes; E correlation between six core gene, red for positive correlation, green represents negative correlation; F six core gene predict disease of ROC curve; "X" 1 – specific degree, Y axis as the "sensitivity".

FMOD was significantly upregulated in the experimental group (p < 0.01), while SBSPON expression decreased (p < 0.05) and SPTB expression decreased significantly (p < 0.01), consistent with expectations(Fig 8). However, ADAMTS1 and S100A12 showed no significant differences between the two groups. These results suggest that hub genes may play critical roles in diabetes-related kidney injury, offering new insights into the molecular mechanisms of DKD and identifying potential therapeutic targets.

## 2.9. Correlation and ROC analysis of hub genes

The correlations of the six hub genes are shown in Fig 7E. Analyzing the ROC curve and evaluating the AUC values of the six hub genes for their diagnostic efficiency in DKD (Fig 7F), we found that EXT1, ADAMTS1, and SPTB showed high diagnostic accuracy, with AUC values of 0.85, 0.82, and 0.838, respectively. S100A12, SBSPON, and FMOD had AUC values of 0.773, 0.769, and 0.802, respectively, which, while lower, still indicate certain diagnostic performance. These results provide new potential biomarkers for the early diagnosis of DKD and lay the foundation for further clinical application studies.

## 2.10. GSEA enrichment analysis of hub genes

We conducted correlation analysis between the 6 hub genes and all other genes, displaying the expression patterns of the top 50 positively correlated genes for each hub gene using a heatmap (Fig 9). Based on the correlation analysis results, we performed GSEA on each hub gene and displayed the top 20 enriched Reactome pathways (Fig 10). According to the GSEA results, pathways with enrichment scores greater than 0 are positively correlated with the genes, while pathways with scores less than 0 are negatively correlated with the genes.

## 2.11. Immune cell infiltration analysis

CD45, which is a classic cell marker, is widely expressed on immune cells. Immunohistochemical analysis demonstrated a significant increase in CD45-positive cells within the renal tissues of the DKD group compared to the control group, suggesting substantial immune cell infiltration in the DKD cohort(Fig 11).

This study used the ssGSEA algorithm to evaluate the degree of immune cell infiltration. We demonstrated the correlation between the proportions of immune cell infiltration (Fig 12A) and compared the differences in immune cell infiltration between the patient group and the healthy control group (Fig 12B). The results show that, compared with the healthy group, the infiltration of activated B cells, gamma delta T cells, natural killer cells, neutrophils, and regulatory T cells increased significantly in DKD patients. Finally, we used the Pearson correlation coefficient to further explore the correlation between immune cells, immune function, and the expression of key genes (Fig 12C). The size of the circle in the figure indicates the strength of the correlation coefficient, and the color represents the p-value. Only immune cells with p < 0.05 are shown.

## 2.12. Prediction of transcription factors and MicroRNAs associated with hub genes

We used the RegNetwork database (https://regnetworkweb.org/) to predict microRNAs (miRNAs) and transcription factors upstream of the six hub genes. A gene regulatory network was then constructed using Cytoscape software (Fig 13).

## 3. Discussion

This study utilized various bioinformatics and machine learning methods to conduct an in-depth analysis of DKD samples, exploring the crucial role of protein glycosylation in the pathogenesis of DKD at the genetic level. The research first obtained datasets from the GEO and MsigDB databases, which were then integrated after data preprocessing. Differential expression analysis and functional enrichment analysis identified a set of differentially expressed genes (DEGs) between DKD and normal kidney tissues. GO and KEGG pathway enrichment analyses revealed that these DEGs were

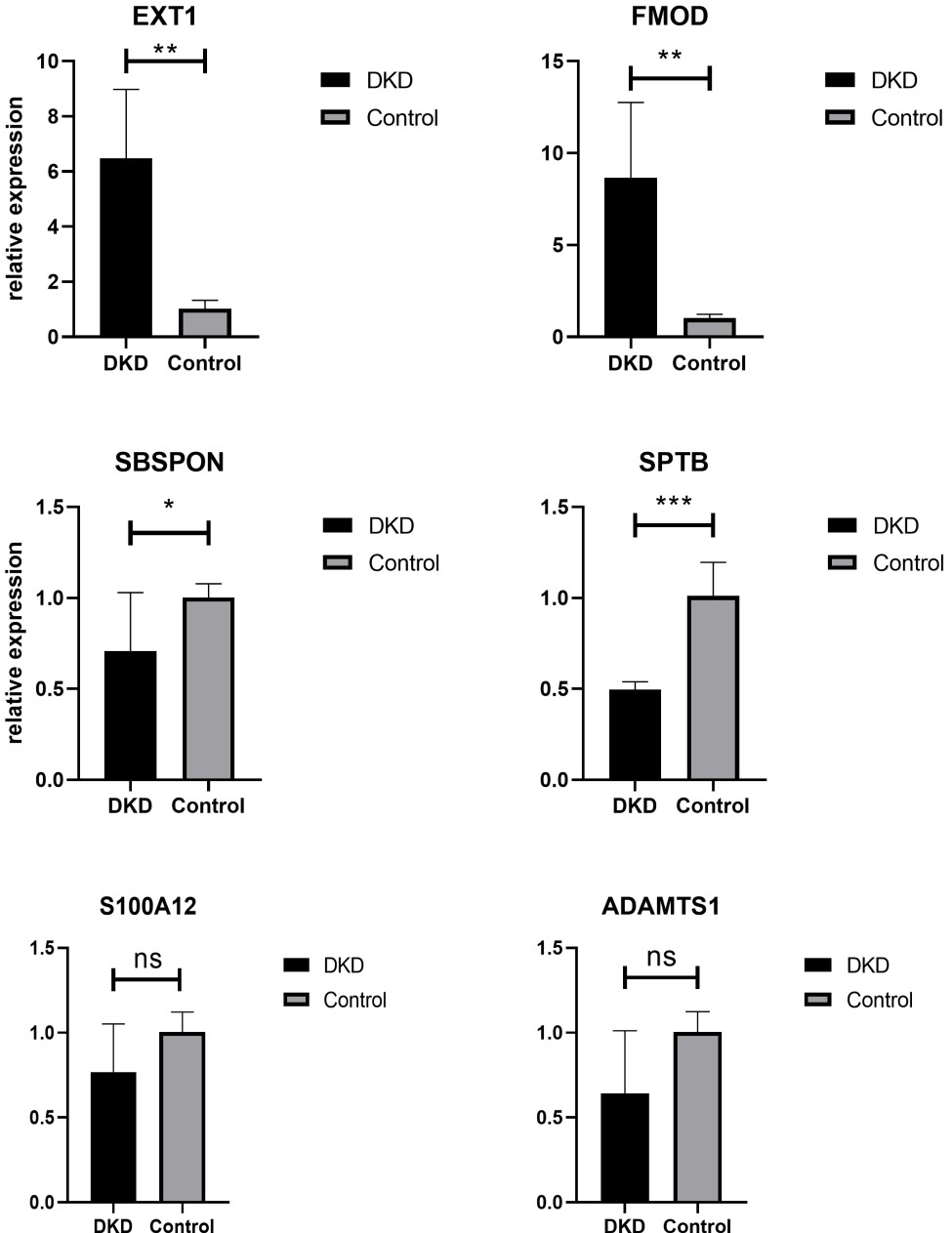

**Fig 8. Relative expression of hub genes.** qPCR results showed that the expression of EXT1 and FMOD was significantly upregulated in DKD mouse($p < 0.01$), while SBSPON expression decreased ($p < 0.05$) and SPTB expression decreased significantly ($p < 0.01$).

mainly enriched in biological processes, cellular components, and molecular functions related to tissue damage, chronic inflammation, renal dysfunction, and the formation of glomerulosclerosis and fibrosis in DKD. Furthermore, these genes were significantly enriched in pathways such as the PI3K-Akt signaling pathway, cytokine-cytokine receptor interaction, and MAPK signaling pathway, which are closely associated with cell proliferation, survival, metabolism, and inflammatory response. This suggests that the abnormal activation of these pathways may be key factors contributing to the pathological progression of DKD.

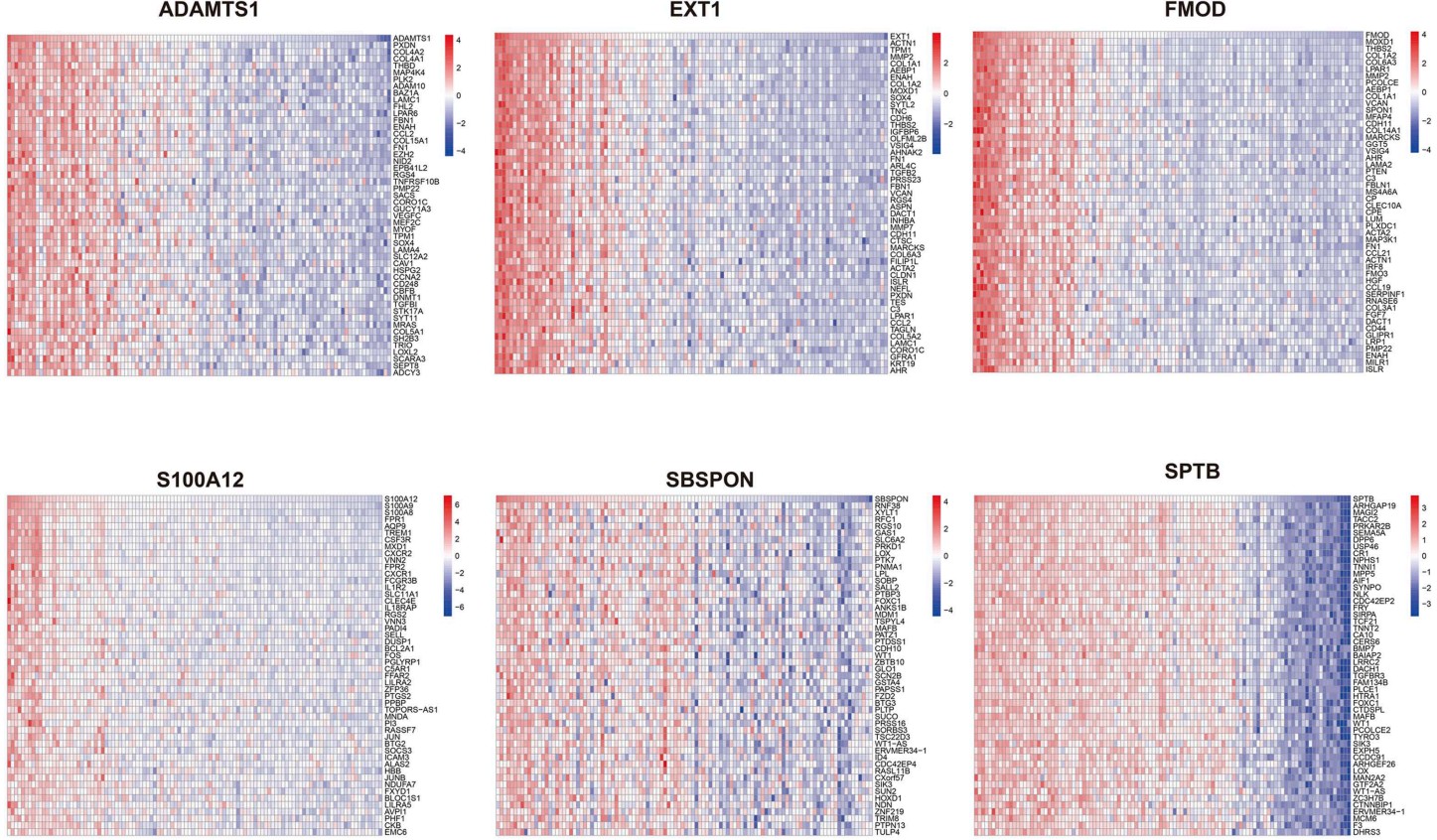

**Fig 9. The correlation analysis between hub genes and all genes.** Conduct a correlation analysis between 6 hub genes and all genes, and respectively display the expression of the top 50 positively correlated genes using heatmaps.

Subsequently, by intersecting DEGs with the glycosylation gene set, 21 gly-DEGs related to glycosylation modification were identified, including 13 upregulated genes and 8 downregulated genes. Further GO and KEGG pathway enrichment analysis revealed that these gly-DEGs were mainly enriched in pathways associated with extracellular matrix organization, extracellular structural organization, and external encapsulating structure organization biological processes. This suggests that they may play an important role in the occurrence and development of DKD by affecting the formation of glomerular sclerosis and fibrosis, the remodeling process of kidney tissue, and the repair and regeneration capacity of the kidney. Additionally, KEGG pathway analysis uncovered close associations between these gly-DEGs and pathways such as autophagy, O-glycan biosynthesis, glycolipid biosynthesis, and heparan sulfate biosynthesis. These abnormal glycosylation modifications may lead to protein dysfunction, which can further impact the pathological progression of DKD. These findings not only reveal the potential mechanisms of action for gly-DEGs in DKD but also provide new research directions for screening therapeutic targets and biomarkers.

Following this, we employed unsupervised clustering analysis to divide the samples into two subtypes and evaluated the differences in pathway activity between these subtypes. The results revealed significant biological differences, which are crucial for an in-depth understanding of the pathogenesis of DKD. By uncovering these differences, we can further explore the unique pathological processes and potential molecular mechanisms associated with each subtype. This approach provides insights into the biological variations among different patients and the diversity in disease progression pathways, thereby driving advancements in precise diagnosis and personalized treatment for DKD.

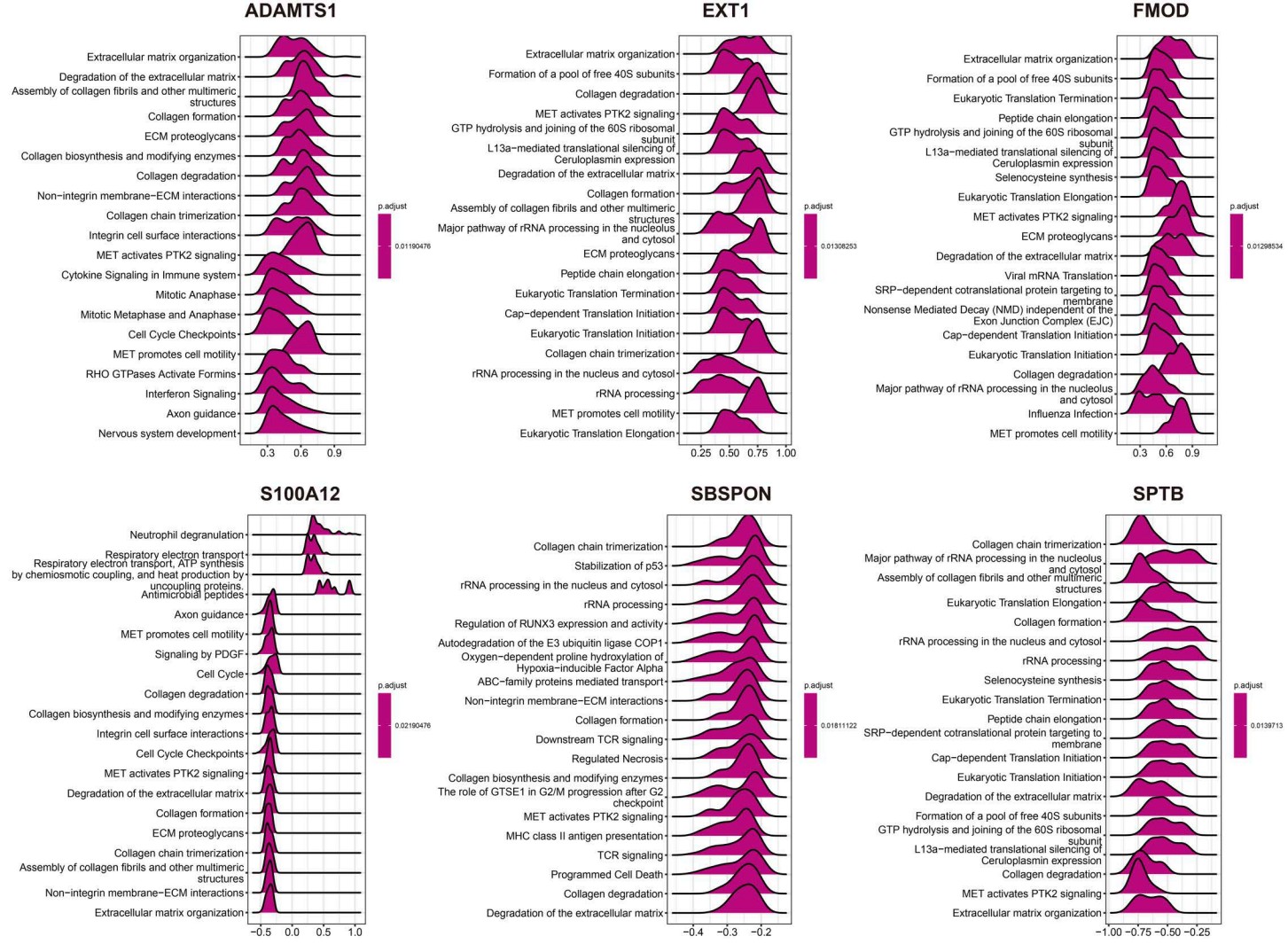

**Fig 10. The top 20 results of the GSEA analysis of the 6 hub genes in Reactome.** The values below represent the enrichment scores, with values greater than 0 indicating a positive correlation between the gene and the pathway, and values less than 0 indicating a negative correlation.

To identify the genes with significant characteristics, various machine learning methods, including LASSO regression, random forests, and support vector machine (SVM), were used in the study. By integrating the results of these three machine learning models and taking their intersection, six hub genes were identified: S100A12, EXT1, SBSPON, ADAMTS1, FMOD, and SPTB. The mechanism of action and potential clinical application value of these genes in the early diagnosis and treatment of DKD provide important clues. In-depth research on the role and potential applications of these genes in the pathogenesis of DKD is essential for developing early diagnostic and treatment strategies.

S100 Calcium Binding Protein A12 (S100A12) encodes a calcium-binding protein that belongs to the S100 protein family and is considered a highly sensitive and specific diagnostic biomarker for local inflammatory reactions, plays an important role in various inflammatory, metabolic, and tumorous diseases [20–22]. Gawdzik, Joseph et al. and Belmokhtar, Karim et al. confirmed that S100A12 accelerates vascular damage in CKD by mediating vascular calcification and remodeling [23,24]. Yayar, Ozlem et al. and Zakiyanov, Oskar et al. confirmed a positive correlation between elevated levels of

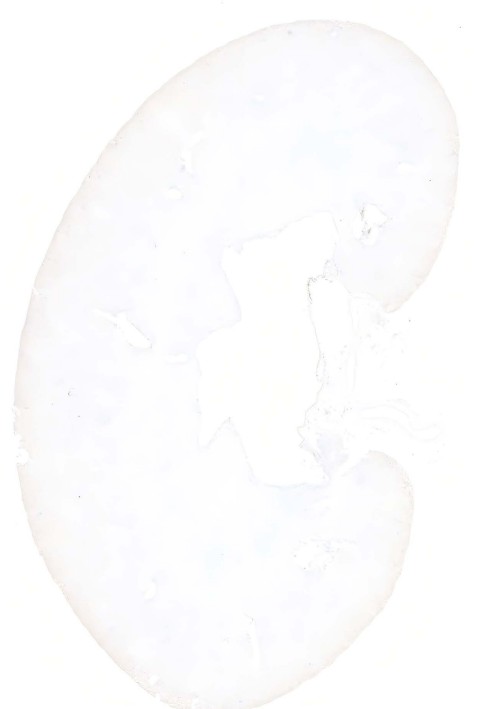
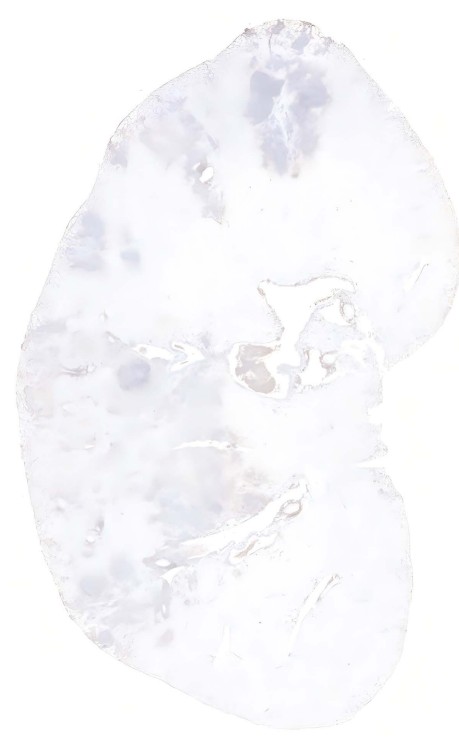

1000 µm

**Fig 11. The kidney IHC results of db/db and db/m mouse.** IHC results showed that the kidney of DKD group exhibited significant immune cell infiltration compared to the control group.

S100A12 in serum and both the occurrence and severity of atherosclerosis [25,26]. Furthermore, the evidence provided by Isoyama, Naohito et al. and Zakiyanov, Oskar et al., respectively, showed that the level of S100A12 in plasma and urine was significantly increased in CKD patients and was positively correlated with serum creatinine levels [27]. These findings suggest that S100A12 not only plays an important role in the development and progression of DKD but also may reflect the status of kidney function.

Exostosin-1 (EXT1) encodes the EXT1 protein, a member of the exostosin glycosyltransferase family, which is co-localized with EXT2 on the glomerular basement membrane (GBM) and play an important role in the pathological processes of the glomerular filtration barrier. Additionally, EXT1 can influence multiple signaling pathways such as FGF, BMP, and Wnt, ultimately leading to abnormalities in GBM and proteinuria. As a recently discovered hotspot, EXT1 has been confirmed to be widely expressed in various membranous nephropathies, including lupus membranous nephropathy (LMN), primary membranous nephropathy (PMN), and PLA2R-negative membranous nephropathy and indicates a favorable prognosis [28–34]. Using type 2 diabetes model constructed by db/db mice, Sampei, So, et al. revealed that EXT1 may maintain vascular function by promoting the synthesis of the vascular endothelial glycocalyx, decrease the risk of vascular damage and inflammatory reactions [35]. Considering the crucial role of EXT1 in the glomerular basement membrane, further investigation into its specific functions will contribute to a more comprehensive understanding of the pathogenesis of DKD.

SBSPON, encoded by the gene Somatomedin B and Thrombospondin Type 1 Domain Containing (SBSPON), is a secreted protein which is predicted to play an important role in metabolic pathways by performing functions such as post-translational modification, including O-linked glycosylation. Although there is limited research on the SBSPON protein currently, numerous studies have confirmed that Thrombospondin-1 protein (TSP1 protein) plays a crucial role in promoting inflammatory responses and oxidative stress through regulating cell-matrix and cell-cell interactions, inducing

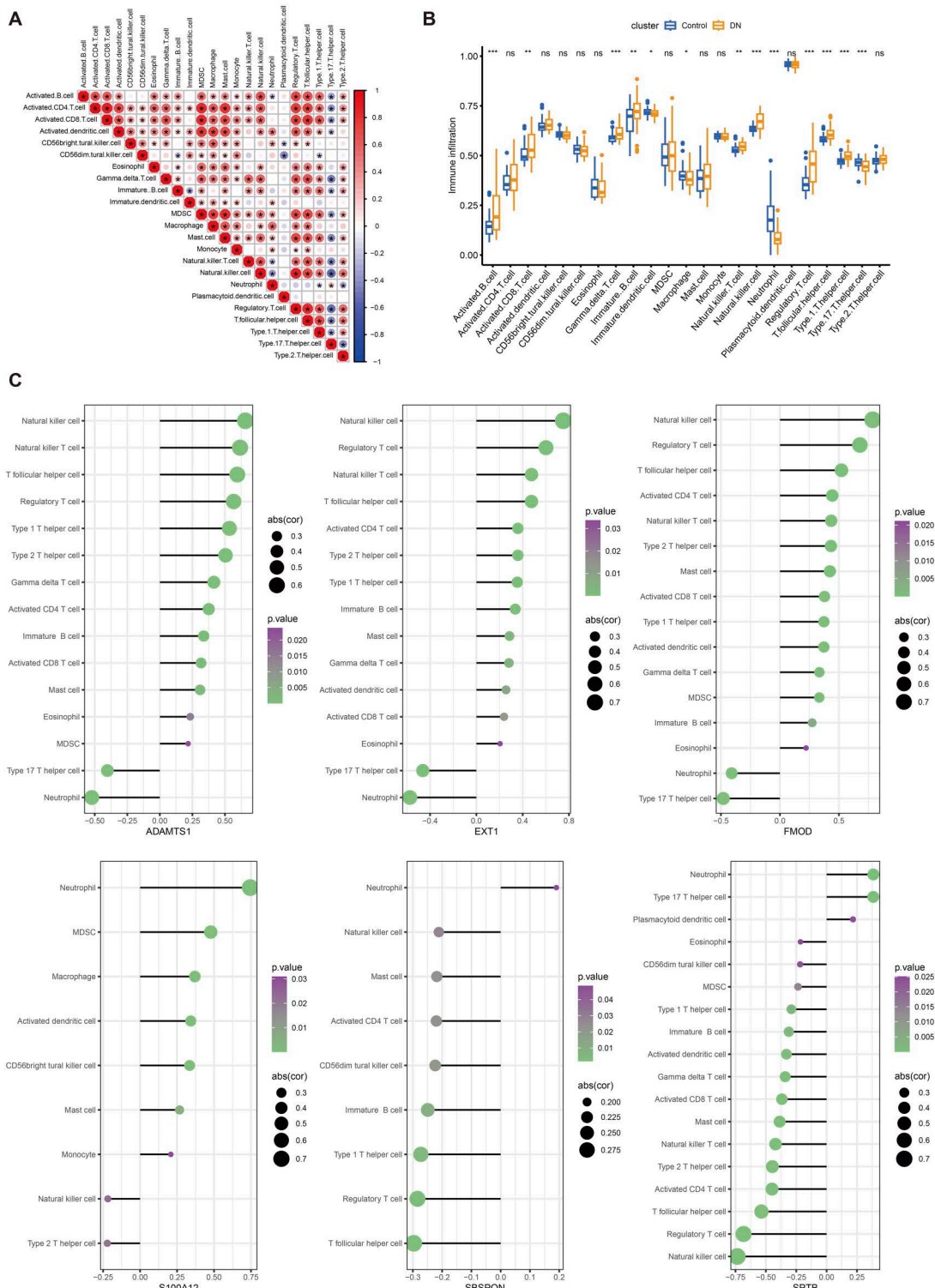

**Fig 12. Immune cell infiltration degree of evaluation.** A shows the correlation between the proportion of the immune cell infiltration, B show patients and healthy controls immune cells infiltrating differences between two groups; Ns (p > 0.05, * p < 0.05, * * * * * p < 0.01, p < 0.001). C, respectively, show six core genes and immune cell infiltration (only show the relevance of the p < 0.05 immune cells). Circle size according to the correlation coefficient, color represents p values.

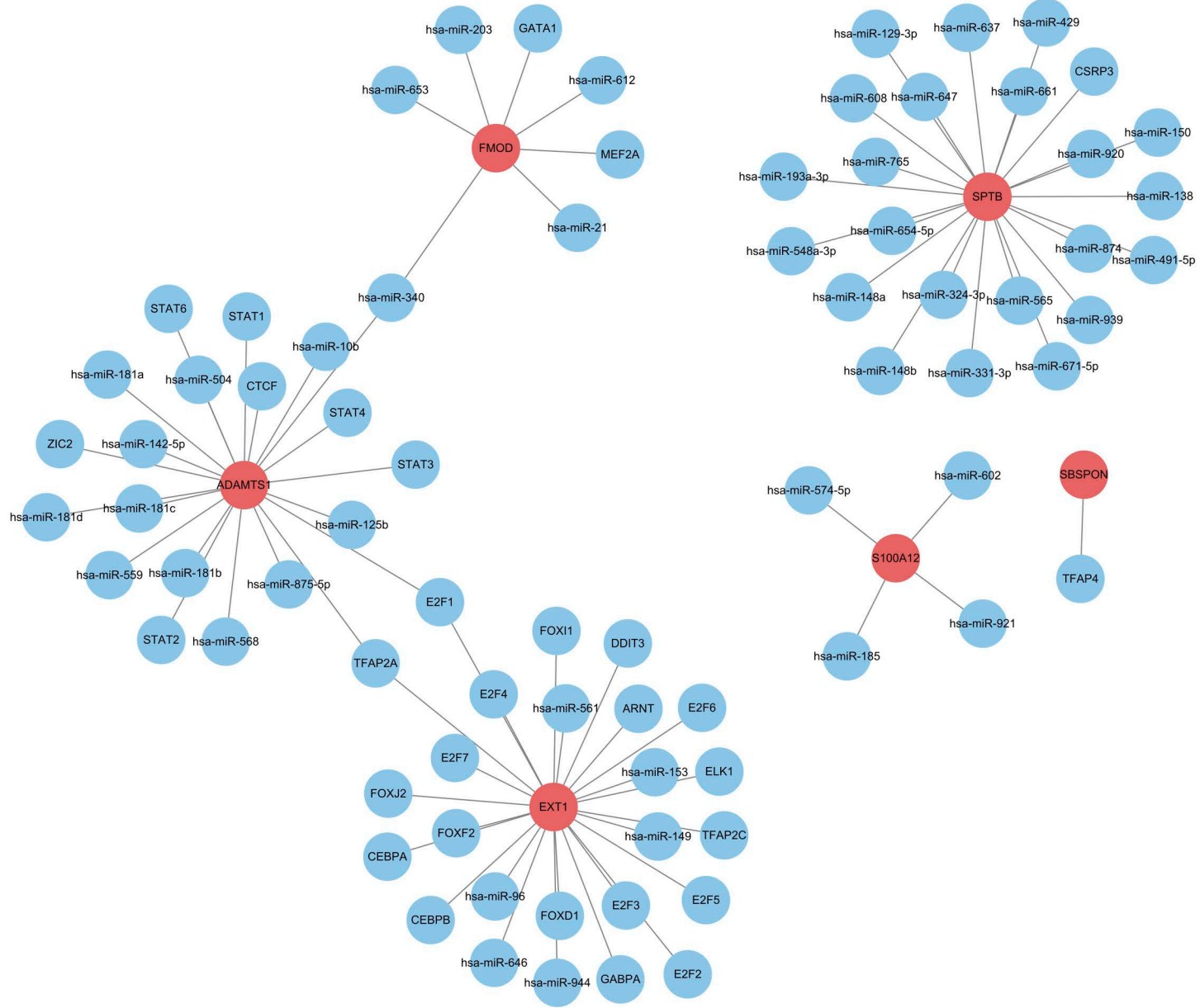

**Fig 13. Prediction of miRNAs and transcription factors upstream of core genes using regnetwork database.** The core genes are in red (only the genes in the database are shown).

apoptosis in endothelial cells and podocytes, inhibiting cell migration and angiogenesis, among other biological processes [36–40]. Additionally, Varma, Vijayalakshmi et al. and Matsuo, Yoshiyuki et al. found that TSP1 is upregulated in obesity and metabolic syndrome and is associated with pathological processes such as insulin resistance and adipose tissue inflammation [41,42]. Considering that both SBSPON and TSP-1 conduct their biological functions through the Thrombospondin-1 domain, SBSPON may play a crucial role in the pathogenesis of DKD by regulating metabolism, inflammatory responses and oxidative stress.

The ADAMTS-1 protein, encoded by A Disintegrin and Metalloproteinase with Thrombospondin Motifs 1 (ADAMTS-1), is a novel metalloproteinase belonging to a special type within the family of A Disintegrins and Metalloproteinases (ADAM). Despite being a secreted protein, it does not exist in the extracellular fluid as a soluble form but rather tightly binds to the extracellular matrix (ECM) through its three thrombomodulin (TSP) type I motifs located at the carboxyl terminus. Additionally, the ADAMTS-1 protein may play an important role in glycosylation modification through its four potential N-linked glycosylation sites [43,44]. Extensive research has demonstrated that ADAMTS-1 plays a crucial role in development, function maintenance, and injury repair of kidney [45–47]. Specifically, ADAMTS-1 promoting renal fibrosis through affecting vascular stability, collagen deposition and ECM modulation, which is crucial in DKD progression [48–51].

The FMOD protein, encoded by Fibromodulin (FMOD), belongs to the Small Leucine-Rich Proteoglycans (SLRP) family class II member and participates in a variety of biological processes in kidney. FMOD is overexpressed in the renal cortex and its protein is found to accumulate significantly in the renal tubular interstitium in DKD [52,53]. In kidney diseases, the expression pattern of FMOD is closely associated with the fibrosis process [54]. In DKD, the deposition of FMOD is also concentrated in fibrotic scar areas, and its expression significantly increases with the severity of fibrosis [55]. Jeansson, M et al revealed that FMOD can bind to activated TGF-β1 and inhibit its expression, thereby delaying the fibrotic process and maintaining the glomerular negative charge barrier. Jazi, Maryam Foroutan et al. revealed that in a streptozotocin-induced DKD rat model, supplementation of FMOD could effectively inhibit the expression of TGF-β1 and significantly alleviate the pathological features of DKD, such as reduced urinary albumin excretion and alleviated glomerular injury [56]. Since FMOD may not be retained at the site of formation but excreted into the blood or urine, detecting its concentration can help explain its accumulation in the body. Previous studies have found that FMOD can serve as a biomarker for DKD present in plasma [57]. Therefore, FMOD has great potential in the diagnosis and treatment of DKD.

The SPTB protein, encoded by the Spectrin beta, erythrocytic (SPTB) gene, belongs to the spectrin gene family and plays an important role in the stability of the erythrocyte membrane. Meglic, Anamarija et al reported an association between increased spherocytic red blood cells related to SPTB and kidney failure associated with UMOD mutations in a three-generation family. Two individuals in the family developed end-stage renal disease, one was in CKD stage 4 accompanied by interstitial fibrosis and tubular atrophy [58]. In addition, Tian, Weijie et al reported SPTB variant patients also presented with left ectopic kidney and adolescent idiopathic scoliosis [59]. These findings suggest that SPTB may important in kidney development.

From what has been discussed above, the six hub genes are essential in the pathogenesis of DKD, as well as, potential biomarkers for the diagnosis and therapeutic of DKD. Therefore, we utilized db/db mice to establish DKD model, compared the relative expression of hub genes. The results showed EXT1, FMOD and SPTB has significant differences between DKD and normal kidney tissue. While our study identified six hub genes associated with DKD pathogenesis, discrepancies were observed between our findings and prior clinical studies. Specifically, S100A12 and ADAMTS1 showed no significant differential expression in the db/db mouse model despite their reported upregulation in human DKD cohorts. This inconsistency may arise from limitations in our animal model, including the small sample size (n = 3 per group), which reduces statistical power to detect subtle changes, and species-specific regulatory mechanisms inherent to murine models. Additionally, disease stage heterogeneity (e.g., early vs. advanced DKD) and transient expression dynamics during pathological progression could mask gene activity. Through ROC curve analysis and AUC value assessment, we systematically evaluated the diagnostic performance of candidate biomarkers in DKD. The results demonstrated that EXT1, SPTB, ADAMTS1, and FMOD exhibited superior diagnostic accuracy with higher AUC values. EXT1, a key enzyme in heparan sulfate biosynthesis, demonstrated robust diagnostic potential (AUC = 0.92). Future studies should explore its role in extracellular matrix remodeling and therapeutic targeting. Although S100A12 and SBSPON showed relatively lower AUC values, they still maintained appreciable diagnostic potential for DKD identification. These results further confirm the value of hub genes as novel biomarkers for the diagnosis, treatment, and prognostic evaluation of DKD.

In addition, this study evaluated the enrichment of hub genes in relevant pathways and the degree of immune cell infiltration through GSEA enrichment analysis and immune analysis. The study found that in DKD, there was a significant increase in infiltrations of activated B cells, γδ T cells, natural killer cells, neutrophils, regulatory T cells, and other immune cells, which further supported the important role of glycosylation modification in the pathogenesis of DKD.

Finally, by predicting the regulatory molecules of hub genes and constructing a gene regulatory network, this study revealed the miRNAs and transcription factors upstream of the hub genes, further elucidating the complex mechanism of gene regulation. These findings not only provide new perspectives for understanding the mechanisms of DKD but also offer valuable references for future therapeutic strategy development.

This study pioneers a novel molecular subtyping framework for DKD by integrating glycosylation-related genes through bioinformatics and machine learning approaches [60]. Unsupervised clustering identified two distinct DKD subtypes that resolve disease heterogeneity, enabling personalized therapies such as antifibrotic agents for ECM-dominant patients. The identified hub genes demonstrated enhanced diagnostic performance (AUC > 0.7) compared to conventional biomarkers. Mechanistically, we revealed glycosylation dysregulation drives macrophage/neutrophil activation, establishing a "glycosylation-immune crosstalk" axis. These findings reframe DKD pathogenesis as a glycosylation-mediated imbalance between extracellular matrix dynamics and immune regulation, providing precise therapeutic targets for precision medicine.

However, our study also has some limitations that need to be considered. Firstly, we did not investigate the specific molecular mechanisms of glycosylation-related genes in DKD. Secondly, all the data provided in this paper are based on public datasets and have only been validated in mice. Further validation is needed in clinical cohorts.

## Conclusion

This study investigates the key role of protein glycosylation in the pathogenesis of DKD by integrating bioinformatics and machine learning techniques. The results suggest that glycosylation modification plays a crucial role in the occurrence and progression of DKD by influencing glomerulosclerosis, fibrosis, immune cell infiltration, and the abnormal activation of related signaling pathways. Six hub genes (S100A12, EXT1, SBSPON, ADAMTS1, FMOD, and SPTB) were identified as promising diagnostic and prognostic biomarkers. These hub genes and associated pathways not only contribute to understanding the pathogenesis of DKD but also provide potential targets for developing new therapeutic strategies to improve management and prognosis for patients with DKD.

## Supporting information

**S1 File. Raw qPCR Ct values.**
(XLSX)

**S2 File. Merge_DN.**
(TXT)

## Acknowledgments

We extend our heartfelt gratitude to all the researchers and technical staff who contributed their expertise and time to this study. GPT 4o was utilized for translation and linguistic refinement in the preparation of this manuscript.

## Author contributions

**Conceptualization:** Ziyang Liu, Yue Gu, Fengmin Shao.

**Data curation:** Ziyang Liu, Wenxin Bai.

**Formal analysis:** Ziyang Liu, Zengyuan Qin, Chunling Huang.

**Funding acquisition:** Yue Gu, Fengmin Shao.

**Methodology:** Ziyang Liu, Na Li.

**Project administration:** Lei Yan.

**Supervision:** Yue Gu, Fengmin Shao.

**Validation:** Ziyang Liu, Zengyuan Qin, Wenxin Bai, Shasha Wang.

**Visualization:** Ziyang Liu, Zengyuan Qin.

**Writing – original draft:** Ziyang Liu.

**Writing – review & editing:** Ziyang Liu.

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
